# Pollution and Risk Assessments of Heavy Metal(loid)s in the Soil around Lead-Zinc Smelteries via Data Integration Analysis

**DOI:** 10.3390/ijerph19159698

**Published:** 2022-08-06

**Authors:** Ziruo Zhou, Chi Peng, Xu Liu, Zhichao Jiang, Zhaohui Guo, Xiyuan Xiao

**Affiliations:** School of Metallurgy and Environment, Central South University, Changsha 410083, China

**Keywords:** smelting sites, heavy metals, meta-analysis, land use, risk assessment, soil pollution

## Abstract

Pb–Zn smelting is a major cause of heavy metal(loid) contaminations in soils. We collected data on heavy metal(loid)s in the soils near Pb–Zn smelteries globally from 54 peer-reviewed reports to study the metals’ distribution, pollution index, and potential ecological and health risks. We observed that 90% of the studied Pb–Zn smelteries were distributed in Asia and Europe. Heavy metal(loid)s were mainly deposited within a 2 km distance to the smelteries, with mean concentrations (mg/kg) of 208.3 for As, 26.6 for Cd, 191.8 for Cu, 4192.6 for Pb, and 4187.7 for Zn, respectively. Cd and Pb concentrations in the soil exceeded their corresponding upper continental crust values several hundred folds, suggesting severe contamination. The smelting area had the highest heavy metal(loid) contamination in soil, followed by the forest land, farmland, and living area. Compared with the soil environmental standard values from various countries, As, Cd, Pb, and Zn were considered priority pollutants for protecting the ecosystem and human health. Likewise, As, Cd, and Pb were suggested as the priority pollutants for protecting groundwater safety. The potential ecological and health risks of heavy metal(loid)s in the soil within 2 km of Pb–Zn smelteries were severe and should be of concern.

## 1. Introduction

Smelting activities cause heavy metal(loid) contamination in soils [1]. Hydrometallurgy accounts for 80% of global Zn production, while most Pb is produced by pyrometallurgy [2,3]. The processes of hydrometallurgy and pyrometallurgy (such as roasting, leaching, and neutralization replacement) produce a large amount of heavy metal(loid)-containing waste gas, water, and slag [4]. After discharge, heavy metal(loid)s are transported by airflow, rainwater leaching, and surface runoff, causing severe pollution to the soil within and around the smelteries [5,6]. For example, China reported that 75% of the soil around smelteries was heavily polluted with heavy metal(loid)s [7]. Excessive heavy metal(loid)s in the soil threaten the health of the ecosystem and humans through the food chain [8]. Because of the upgrading environmental standards, many smelting sites in the suburbs have relocated, leaving behind severely contaminated soils [9]. Studying the status, distribution, and risk of heavy metal(loid)s in the soil around Pb–Zn smelteries is of great significance to the safe use of the land.

The distribution of heavy metal(loid)s in the soil around a smelter is affected by many factors. Often, the heavy metal(loid) concentration correlates negatively with the distance to the production area of the smelter [6]. In addition to distance, land-use type (strongly correlated with the heavy metal(loid) source, the intensity of human activities, and soil properties) is the primary factor affecting heavy metal(loid) concentrations in soils [10]. Research shows that As concentrations in construction land are higher than in forest land and farmland because human activities are more active in construction lands [11]. Wu et al. [12] found that the farmland in a Pb–Zn mine was generally polluted because of wastewater and excessive fertilizers. Furthermore, land use affects soil pH, cation exchange capacity (CEC), and organic matter, further influencing the accumulation and distribution of heavy metal(loid)s [13]. Soils in forest land are less affected by human activities but have a high organic matter concentration, which may enrich heavy metal(loid)s [14]. Knowledge of the relationship between land use and heavy metal(loid) concentration distribution in the soil around smelteries would aid soil pollution prevention and remediation.

The pollution index (PI) is widely used to assess the pollution level of heavy metal(loid)s in soil. It assesses pollution extent based on the soil environmental standard values (SSVs) or soil background values for heavy metal(loid)s as reference values. Due to differences in policies, geographical conditions, and methodologies, the SSVs may differ between countries [15]. Sun et al. [16] found that the difference in toxicity assessment and risk characterization methods of carcinogens results are the main reasons for the significant differences in SSV between countries. Meantime, the SSVs vary with land-use types [17]. Many countries have issued different SSVs for different land-use types because the risk receptors and exposure pathways in the assessment models vary with land use. In addition to SSVs, soil background values are popular reference values that also vary spatially [18]. Therefore, choosing the reference values in assessing soil pollution requires diligence.

Risk assessment methods, such as the ecological risk and health risk assessment for heavy metal(loid)s in soil, are widely used to estimate soil pollution. For instance, Hakanson potential ecological risk (PER) index assesses heavy metal(loid)s’ impacts on the ecological system based on their toxicity and concentration [19]. For example, Shen et al. [1] found that the potential ecological risk of Cd in a Pb–Zn smelter in China was very high. The health risk assesses heavy metal(loid)s’ risk on human health by evaluating their exposure route and exposure frequency [20]. The heavy metal(loid)s in the soil of industry districts would pose a carcinogenic and non-carcinogenic risk to residents [11]. Because the landscape and the environment of smelteries are complex, combining multiple assessment methods to evaluate heavy metal(loid) pollution would help avoid bias [21,22].

Many case studies focused on a single smelting site have reported on the contamination of heavy metal(loid)s in the soil around Pb–Zn smelteries [23,24,25]. Because each smelter has a peculiar smelting process and geographical environment, these case studies are difficult to generalize heavy metal(loid)s distributions in the soil around the Pb–Zn smelteries. A literature review reported the high risk posed by Pb in the soil near Pb–Zn smelteries in China [26]. Currently, there is a lack of study on heavy metals in the soil around smelteries on a global scale. Therefore, we gathered data from published articles on a global scale, and compared them with SSVs from different countries, to (1) study the characteristics and distribution of heavy metal(loid)s in soils with various land uses around Pb–Zn smelteries and (2), identify the essential risk elements and risk types by assessing the pollution index, potential ecological risk, and health risk of heavy metal(loid)s in the soil. The results would provide suggestions for pollution identification and risk prevention of heavy metals in smelting-affected areas.

## 2. Materials and Methods

### 2.1. Data Collection

The As, Cd, Cu, Pb, and Zn concentration data in soils around Pb–Zn smelteries were collected by screening published peer-viewed articles between 1997 and 2019. These articles were searched from peer-viewed literature databases: Web of Science, Science Direct, and China National Knowledge Infrastructure (CNKI) using keywords such as Pb–Zn smelting, heavy metal(loid)s, soil, and land use. The articles were further screened according to several criteria, including the number of sampling sites larger than 3, soil sampling depth less than 30 cm, sampling radius less than 20 km from the smelteries, and provided the statistical values of heavy metal(loid) concentrations. Overall, 54 relevant peer-viewed pieces of literature and the measurement data of 3547 soil samples from 18 countries around the world were obtained after the screening.

The collected data were divided into two groups according to the sampling radius (Table 1 and Appendix A). The data in Group #1 were those from soil samples within a 2 km radius of Pb–Zn smelteries, while the data in Group #2 were those between 2 and 20 km from the smelteries. The sampling sites were classified into four land uses: forest land, farmland, smelting area (smelting production area), and living area. All of the adopted articles used as data sources in this study are listed in the Appendix A.

### 2.2. Geo-Accumulation Index (I_geo_)

*I*_geo_ is a widely used method for assessing the impact of human activities on heavy metal(loid)s in soil [27]. The *I*_geo_ values were calculated by using Equation (1):*I*_geo_ = log_2_[*C*_i_/(1.5 × *B*_i_)](1)
where *C*_i_ is the measured concentration (mg/kg) of As, Cd, Cu, Pb, or Zn in soil. *B*_i_ (mg/kg) is the reference value of heavy metal(loid) i. *B*_i_ values were set to the upper continental crust values (UCC) (mg/kg) in the current study, which are 5.7, 0.06, 27, 25, and 75 for As, Cd, Cu, Pb, and Zn, respectively [28]. The classes of *I*_geo_ values for identifying the pollution level of heavy metal(loid)s in the soils are listed in Appendix A.

### 2.3. Nemerow Integrated Pollution Index (NIPI)

NIPI and PI were adopted to evaluate the pollution level of multiple pollutants in soils [29]. NIPI and PI were calculated as follows:(2)$$\mathrm{NIPI}=\sqrt{\left({{\mathrm{PI}}_{\max}^{b}}^{2}+{\bar{{\mathrm{PI}}^{b}}}^{2}\right)/2}$$
(3)$${\mathrm{PI}}_{i}^{b}=C_{i}/B_{i}$$
(4)$${\mathrm{PI}}_{i}^{s}=C_{i}/S_{i}$$
where ${\mathrm{PI}}_{i}^{b}$ is the pollution Index based on the soil background value. ${\mathrm{PI}}_{\max}^{b}$ is the maximum value of${\;\mathrm{PI}}_{i}^{b}$, $\bar{{\mathrm{PI}}^{b}}$ is the average value of ${\mathrm{PI}}_{i}^{b}$. *C*_i_ is the concentration (mg/kg) of heavy metal(loid) (pollutant) i in the soils, *B*_i_ is the reference value for heavy metal(loid)s, which can be set to background value or the soil environmental standard value$,{\;\mathrm{PI}}_{i}^{s}$ is the pollution Index based on the soil environmental standard value, and *S*_i_ is the SSVs of the heavy metal(loid) pollutant i. The sources of SSVs are shown in Appendix A. Appendix A lists the pollution levels according to the NIPI and PI values.

### 2.4. Risk Assessment

#### 2.4.1. Potential Ecological Risk (PER)

The PER index, proposed by Ref. [30] was applied to assess the degree of heavy metal(loid) pollution in soils, according to the toxicity of heavy metal(loid)s and environmental response. RI is the comprehensive potential ecological risk index, representing the biological community’s sensitivity to toxic substances. RI indicates the potential ecological risk level caused by the overall pollution. The PER indices are calculated using Equations (5) and (6), respectively.
(5)$$\mathrm{ER}\left(i\right)=T_{r}^{i}\times {\mathrm{PI}}_{\mathrm{bi}}$$
(6)RI=∑ER(i)

The corresponding toxicity coefficients ($T_{r}^{i}$) for As, Cd, Cu, Pb, and Zn are 10, 30, 5, 5, and 1, respectively [31]. See Appendix A for potential ecological risk levels.

#### 2.4.2. Health Risk Assessment

This study adopted the risk-based soil screening levels (SL_s_) to calculate the health risk of heavy metal(loid)s in soil. The SL_s_ were established by the U.S. Environmental Protection Agency (USEPA) to quickly assess the carcinogenic risk (CR) and non-carcinogenic risk (NCR) of specific chemical substances [32]. The SL_s_ values refer to the acceptable pollutant concentration in the soil by considering the exposure of children and adults. The total CR and NCR are calculated as follows:(7)$$\mathrm{Total}\;\mathrm{CR}=\left(\sum \frac{C_{i}}{{\mathrm{SL}}_{i}^{c}}\right)\times 10^{-6}$$
(8)$$\mathrm{Total}\;\mathrm{NCR}=\sum \frac{C_{i}}{{\mathrm{SL}}_{i}^{n}}$$
where *C*_i_ (mg/kg) is the heavy metal(loid) concentration in soils, ${\mathrm{SL}}_{i}^{c}$ (mg/kg) represents the screening level of carcinogens i, and ${\mathrm{SL}}_{i}^{n}$ (mg/kg) is the screening level of non-carcinogens i. The USEPA defines As as a carcinogen and Cd, Cu, Pb, and Zn as non-carcinogens. The SL_s_ for As, Cd, Cu, Pb, and Zn are 0.68, 71, 3100, 400, and 23,000 [33]. CR < 10^−6^ suggests a safe level, CR of 10^−6^~10^−4^ suggests potential carcinogenic risk, and CR ≥ 10^−4^ suggests high carcinogenic risk. NCR values < 1 suggest a safe level, and values ≥ 1 suggest non-carcinogenic risk.

### 2.5. Data Analysis

This study used SPSS Statistics 22.0 software (IBM, New York, NY, USA) to calculate the statistics of the heavy metal(loid) data in soils near Pb–Zn smelteries; *I*_geo_, PI, NIPI, PER, and health risk index were calculated by Excel 2019 (Microsoft, Redmond, WA, USA). Pearson’s correlation analysis was conducted based on the log-transformed data of heavy metal(loid) concentrations in the soil. ArcGIS 10.2 (ESRI, Redland’s, CA, USA), Sigmaplot 14.0 (Corel Corporation, Ottawa, ON, Canada), and GraphPad Prism 9.0 (GraphPad, San Diego, CA, USA) were used to draw the global distribution of the NIPI values, *I*_geo_ boxplot and heavy metal(loid) concentrations histogram.

## 3. Results and Discussions

### 3.1. Characteristics of Heavy Metal(loid)s in Soils near Pb–Zn Smelteries

In Group #1, the mean concentration (mg/kg) of As, Cd, Cu, Pb, and Zn in the soils around the smelteries were 208.3, 26.6, 191.8, 4192.6, and 4187.7, respectively; they were 36.5, 266.1, 7.1, 167.7, and 37.2 times the crustal background value, respectively (Table 1). On average, the *I*_geo_ values of As, Cd, Cu, Pb, and Zn in Group #1 were 3.0, 7.2, 1.1, 4.7, and 3.0, respectively (Figure 1). Cd and Pb accumulated higher than other metals, with Cu found to be the least accumulated. It was suggested that high amounts of Cd and Pb were found in the exhaust gas from the Pb–Zn smelting process, entering the soil by atmospheric deposition [34,35]. Further, the coefficient of variation (CV) reflects the difference in the mean concentrations between studies. The order of CV was Pb (200%) > As (190%) > Zn (170%) > Cu (150%) > Cd (130%) (Table 1), suggesting the data varied largely among the various smelting sites. The differences in the raw materials, smelting processes, environmental facilities, technologies, and local environmental policies may lead to the difference in heavy metal(loid) emissions, which in turn, cause high variability of heavy metal(loid)s in the soil [36,37].

The mean concentrations of As, Cd, Cu, Pb, and Zn in Group #2 were 19.7, 11.6, 96.1, 420.8, and 789.2 mg/kg, respectively, which were 3.5, 116.1, 3.6, 16.8, and 10.5 times of the crustal background values (Table 1). The mean *I*_geo_ values in Group #2 were 0.9 for As, 6.2 for Cd, 1.1 for Cu, 2.7 for Pb, and 2.0 for Zn. It suggested an accumulation of Cd, Pb, and Zn in the soil. Compared with the data of Group #1, the mean concentrations of heavy metal(loid)s in Group #2 were significantly reduced by 2–10 times (Table 1 and Appendix A). The differences in these data between the two groups were caused by the different sampling radius, suggesting the heavy metal(loid)s mainly deposited in the soil close to smelteries.

The proportions of heavy metal(loid)s with *I*_geo_ > 3 in Group #1 were Cd (97.4%) > Pb (73.0%) > Zn (55.6%) > As (38.1%) > Cu (20.1%) (Figure 1). In Group #2, we observed Cd (100%) > Pb (39.1%) > Zn (36.4%) > As and Cu (0%). About 80% of the Pb–Zn smelteries used Pb–Zn sulfide as a raw material [38], divided into sphalerite and galena, the main constituent minerals of Pb and Zn, respectively [39]. Sphalerite contains high Cd in nature leading to the excessive Cd, Pb, and Zn found in the smelting fumes, wastewater, and slags [40]. The high accumulation of Cd, Pb, and Zn in the soil around Pb–Zn smelteries should be prioritized. More specifically, the distribution and risk of heavy metal(loid)s in soil within 2 km of the smelteries deserves further study.

### 3.2. Global Distribution of Heavy Metal(loid)s in Soils around Pb–Zn Smelteries

Figure 2 shows the NIPI values of heavy metal(loid)s in the soil around Pb–Zn smelteries globally. Nearly 58% of the total smelteries studied were distributed in Asia, followed by 32% in Europe, and the remaining 10% of smelteries were in North America, Africa, and South America. The NIPI values varied among countries. Only three sites had an NIPI < 3, while the other 52 sites showed values ranging from 3 to 1980, attributed to polluted soils. Soils from four smelteries (in Slovenia, Brazil, Britain, and Poland) exhibited NIPI values > 1000. The four smelteries had been abandoned for many years (1896–1996) after high production and high waste discharge history [41,42,43,44]. In addition, the high levels of heavy metal(loid)s in soil may be related to the disorderly stacking of Pb–Zn slags in the areas. Approximately 180,000 m^3^ of tailings with high Cd and Pb had been deposited around the smelter in Brazil [42]. Statistically, more than 100 million tons of Pb–Zn slag were stacked in various lands near smelteries in Poland [45].

Ten smelteries had NIPI values ranging from 500 to 1000, of which four were distributed in China, and one was located each in Poland, France, Britain, Bulgaria, Mexico, and the Czech Republic. In China, 42.4% of the smelting sites had NIPI values ranging from 3 to 100. In comparison, only 12.1% of the smelting sites had NIPI exceeding 500. These results indicated that the soils around smelting sites in China were mostly moderately contaminated. China is a major producer of Pb and Zn globally. In 2018, China’s refined Pb and Zn output was 483 × 10^4^ t and 573 × 10^4^ t, accounting for 41.5% and 43.1% of the total global production, respectively [46]. The Pb and Zn smelting industry in China developed late and began with small but numerous production scales [47]. This condition explained the low-medium NIPI values despite the numerous smelting sites in China.

The distribution of PI^b^ values (Appendix A) suggested that the soils around Pb–Zn smelteries were contaminated with Cd, Pb, and Zn in most countries, while As and Cu were less accumulated. PI^b^ value of Cd in the soils around the smelteries was generally higher than that of the other four metals. For example, the PI^b^ of Cd ranged from 1000 to 2000 in Poland and Brazil, while those of As, Pb, and Zn were <500. However, the values found in China (100–500) were lower than in many countries. Such an occurrence may be attributed to the differences in national Pb–Zn ore grade, i.e., 17.1% in China, 9.1% in Brazil, and 5.7% in Poland [48]. The use of low-grade ore in the smelting process would affect the quality of the concentrate, resulting in difficulty in smelting, high energy consumption, and relatively high pollution [49].

Significant positive correlations were found between As, Cd, Cu, Pb, and Zn concentrations in the soil around the Pb–Zn smelteries (Appendix A), indicating that the distributions of these metals were similar and affected by Pb–Zn smelteries’ emissions. Smelteries may simultaneously emit As, Cd, Cu, Pb, and Zn during the production process [50]. For example, the primary pollutants in the exhaust gas during the Pb–Zn smelting are As, Cd, and Pb [51,52]. Moreover, the primary pollutants in the wastewaters are As, Cu, Pb, and Zn [53,54], and those released from smelting slags include As, Cd, Cu, Pb, and Zn [55,56]. These pollutants disperse into the surrounding soil environment through atmospheric deposition and surface runoff, leading to concurrent soil contamination by many heavy metal(loid)s.

### 3.3. Impacts of Land Use on Heavy Metal(loid)s in the Soil around Smelteries

The heavy metal(loid)s in Group #1 (within 2 km to the smelteries) were used to study land use impacts. Generally, the smelting area had the highest heavy metal(loid) concentrations, followed by forest land, farmland, and living area (Figure 3). Specifically, Pb levels in the forest land were higher than in other land use types. High Cu concentrations were found in the smelting area and the forest land. The forest land in the current study is adjacent to the smelter production area. Because of the high canopy density, forest land strongly intercepts the smoke and dust from the smelteries [5]. Second, the forest land has higher soil organic matter, increasing the adsorption of heavy metal(loid)s [57]. The farmland had relatively higher concentrations of As and Zn than other land uses. Characteristically, the farmland is generally located in low terrain areas that are easily affected by sewage irrigation and polluted runoff [58]. The cultivation activities lower the soil pH and increase the electrical conductivity (EC), facilitating the accumulation of As and Zn in farmlands [59]. On the other hand, living areas are far from the smelter, exhibiting the lowest heavy metal(loid) concentrations in the soil. In short, land uses affect the heavy metal(loid)s in soil mainly by altering the transport processes in the soil.

### 3.4. Priority Pollutants Based on National Environmental Standards

To identify the priority pollutants, we collected the SSVs from different countries (Table 2). The SSVs for heavy metal(loid)s in soil varied with land uses and countries. For example, the SSVs from Canada were divided by land use into farmland, industrial, commercial, and living areas, while Belgium strictly divided living areas into recreational areas and living areas with or without vegetable gardens. The difference in protection goals and evaluation methods caused the differences in the SSVs between countries (Appendix A) [15]. China chose the human health risk assessment to establish their SSVs [60], while the U.S. established Ecological Soil Screening Levels (Eco-SSL_s_), which use plants, soil invertebrates, birds, and mammals as risk receptors [61]. Canada has a relatively stringent standard because it calculates for both environmental and human health to establish the SSVs [62]. On the contrary, the SSVs in the Wallonia region of Belgium are relatively loose because they use excessively flexible parameters in the evaluation model, e.g., the groundwater dilution factor was set to 30, while other countries are between 1 and 20 (Appendix A).

The current study used SSVs from different countries to evaluate the PI of heavy metal(loid)s in soils within 2 km of Pb–Zn smelteries (Table 2). A PI > 1 suggests exceeding the standard. The PI values changed mainly because of the difference in SSV magnitude among the countries. Specifically, the Pb and Zn in the smelting area exceeded all of the soil standards. As and Cd in the smelting area exceeded several standard values from the United States, Belgium, and Canada. Furthermore, Cd exceeded all of the soil standards in the farmland while As, Cu, Pb, and Zn exceeded most standards except in Belgium. Similarly, Cd in the living areas exceeded all of the soil standards, while Pb and Zn in the living areas exceeded most of the SSVs, except in Belgium. In forest land, As, Cd, Cu, Pb, and Zn exceeded at least one of the soil standards. By comparing with the SSVs, the PI values of Cu were lower in all types of land use. The results suggested that the priority pollutant in the farmland and living area around Pb–Zn smelteries was Cd, followed by As, Pb, and Zn. Therefore, to protect the ecosystem and human health, As, Cd, Pb, and Zn require priority control because their PI values were >10 (based on China, Canada, and New Zealand standards). Moreover, to ensure groundwater safety, As, Cd, and Pb, whose PI were >10 (based on Belgium and the United States standards), should be prioritized.

### 3.5. Priority Pollutants Based on National Environmental Standards

#### 3.5.1. Potential Ecological Risk Assessment

Table 3 shows the ecological risks (ER) of As, Cd, Cu, Pb, and Zn under different land uses within 2 km of Pb–Zn smelteries. The total potential ecological risks (RI) of smelteries and the surrounding soils are generally high, with the highest RI in the smelting area. The proportion of the sites with high ER followed the trend: smelting area (92.3%) > farmland (85.7%) > forest land (83.3%) > living area (80.1%). The contributions of As, Cd, Cu, Pb, and Zn to RI are 2.7%, 90.7%, 0.3%, 6.1%, and 0.3%, respectively. All smelting sites had a high ER of Cd, while high Ers of Pb were found in 50.0% of the forestland, 33.4% of the farmland, and 53.8% of the smelting area. Only 50.0% of the smelting area and 10.0% of the farmland had a high ER of As in the soil. The potential ER of Cu and Zn in all sites was low, while the high potential ER of Cd and Pb were attributable to their high toxicity and concentrations in the soil. Attention should be paid to the ecological risks caused by Cd and Pb in the soil of smelting areas and farmlands surrounding the smelteries.

#### 3.5.2. Health Risk Assessment

Figure 2 and Table 4 show the health risks of heavy metal(loid)s poses in the soil with different land uses within 2 km of Pb–Zn smelteries. The CR of each land use was >1, indicating that As in soil posed a potential carcinogenic risk. A total of 57.1%, 33.3%, and 30.1% of sites with CR > 100 were found in the smelting area, forest land, and farmland, respectively, indicating that As posed a high carcinogenic risk to the three land uses. The average values of NCR were Pb > Cd > Zn > Cu, in which Pb and Cd were >1, suggesting a non-carcinogenic risk. In terms of land uses, the total risk of Cd, Cu, Pb, and Zn was ranked as follows: forest land > smelting area > farmland > living area. In addition, the proportions of sites with NCR-Pb > 1 is as follows: smelting area (72.2%) > living area (60.1%) > agricultural area (57.1%) > forest land (50.1%). For NCR-Cd > 1, the trend follows: forest land (20.1%) > smelting area (6.7%) > farmland (5.9%).

Consequently, Cu and Zn had low health risks, attributed to their lower toxicity in the soil. We found that the non-carcinogenic risk of Pb in living areas, and Cd and Pb in forest land, farmland, and smelting areas should be of concern. From the perspective of health risk prevention, As, Cd, and Pb in the soils around Pb–Zn smelteries were the prioritized pollutants.

## 4. Conclusions

Data on the concentrations of As, Cd, Cu, Pb, and Zn were collected from published literature in the past twenty years (1997–2019) to study the distribution and risk of heavy metal(loid)s in the soil around Pb–Zn smelteries. Nearly 58% of the studied smelteries were distributed in Asia, followed by Europe (32%), and the rest (10%) in North America, Africa, and South America. The soil around Pb–Zn smelteries was generally contaminated with As, Cd, Cu, Pb, and Zn. The mean concentrations of heavy metal(loid)s in the soils within 2 km of the smelteries were significantly higher than those outside this distance. The soil heavy metal(loid) levels within 2 km of the smelteries followed an order of smelting area > forest land > farmland > living area. SSVs from different countries were used to identify the priority heavy metal(loid)s in the soil around Pb–Zn smelteries, suggesting that As, Cd, Pb, and Zn should be control-prioritized to protect the ecosystem and human health. Meanwhile, As, Cd, and Pb are indicated as the priority pollutants for protecting groundwater safety. The potential ecological and health risks were severe in the soil within 2 km of Pb–Zn smelteries. Cd and Pb were the major contributors to the potential ecological risk, while As, Cd, and Pb were the major contributors to health risks. Soil remediation and risk prevention practices need to focus on As, Cd, and Pb in soils within 2 km of smelteries.

## Figures and Tables

**Figure 1 ijerph-19-09698-f001:**
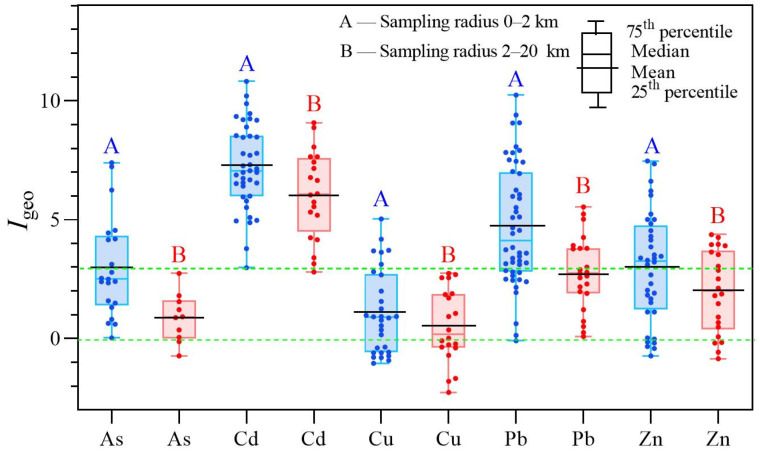
*I*_geo_ values of heavy metal(loid)s in soils near Pb–Zn smelteries globally.

**Figure 2 ijerph-19-09698-f002:**
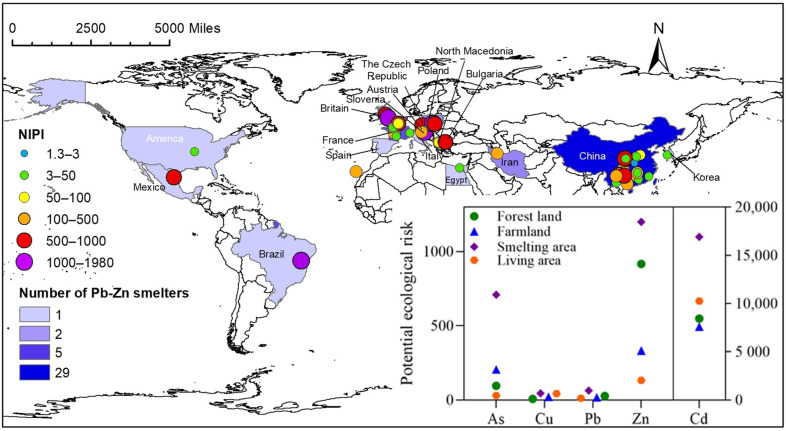
Global distribution of NIPI values of heavy metal(loid)s in soils surrounding Pb–Zn smelteries.

**Figure 3 ijerph-19-09698-f003:**
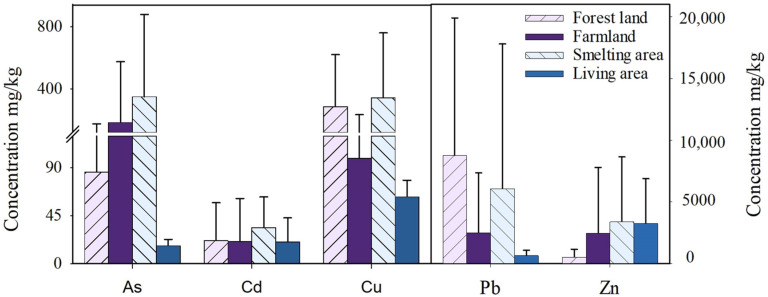
Mean and standard deviation of heavy metal(loid) concentrations in soils within 2 km of the smelteries under different land uses.

**Table 1 ijerph-19-09698-t001:** Statistics of heavy metal(loid) concentrations (mg/kg) in soils near Pb–Zn smelteries globally.

	n	Min	Median	Max	Mean	SD	CV%	UCC ^a^
Sampling radius 0–2 km								
As	22	8.7	46.9	1442.0	208.3	398.9	190	5.7
Cd	39	0.7	12.0	163.0	26.6	33.3	130	0.1
Cu	30	19.7	75.3	1321.3	191.8	285.3	150	27.0
Pb	48	35.2	666.3	45,272.0	4192.6	8242.7	200	25.0
Zn	36	67.9	1074.1	19,859.0	2787.7	4710.9	170	75.0
Sampling radius 2–20 km								
As	10	5.2	16.0	57.1	19.7	15.3	80	5.7
Cd	20	0.6	6.0	48.7	11.6	13.4	120	0.1
Cu	20	8.5	46.5	271.7	96.1	91.1	90	27.0
Pb	23	39.8	250.1	1738.0	420.8	456.9	110	25.0
Zn	22	62.5	449.4	2333.3	789.2	746.6	90	75.0

^a^ UCC refers the upper continental crust values (mg/kg).

**Table 2 ijerph-19-09698-t002:** Soil environment standard values and the corresponding PI values of heavy metals within 2 km of the smelteries.

Country	Standard Value of Soil Environment (mg/kg)					PI for Standard Value				
	As	Cd	Cu	Pb	Zn	As	Cd	Cu	Pb	Zn
	Forest land									
U.S.A.	18	32	70	120	160	4.79	0.69	4.09	73.22	3.41
Belgium	926	2.7	362	581	721	0.09	8.12	0.79	15.12	0.76
	Farmland									
China	40	0.3	50	90	200	4.60	67.57	7.67	27.60	11.57
Belgium	820	12	587	2492	4156	0.22	1.69	0.65	1.00	0.56
Canada	12	1.4	63	70	200	15.35	14.48	6.09	35.48	11.57
Japan	15		125			12.28		3.07		
Czech	65	10	250		1500	2.83	2.03	1.53		1.54
	Smelting area									
China	60	65	18,000	800		5.83	0.56	0.02	8.05	
U.S.A.	1	0.4			620	349.86	91.21			5.97
Belgium	917	19	594	1837	2953	0.38	1.92	0.65	3.51	1.25
Canada	12	22	91	600	360	29.15	1.66	4.22	10.74	10.28
New Zealand	70	13		3300		5.00	0.03		1.95	
	Living area									
China	20	20	2000	400		0.85	1.02	0.03	1.65	
U.S.A.	1	0			620	17.08	51.20			5.27
Belgium	683	10	489	2077	3646	0.03	2.05	0.13	0.32	0.90
Canada	12	10	63	140	200	1.42	2.05	1.00	4.72	16.34
New Zealand	20	3		210		0.85	6.83		3.14	

**Table 3 ijerph-19-09698-t003:** Potential ecological risks of different land-use types in soils within 2 km of the smelteries.

Land Use	ER ^a^-As	ER-Cd	ER-Cu	ER-Pb	ER-Zn	RI
Forest land	151	10,967	53	1757	7	12,935
	(42–326)	(1530–42,425)	(5–136)	(42–5054)	(0.9–22)	
Farmland	323	10,135	17	497	31	11,004
	(15–2256)	(355–81,500)	(4–936)	(7–4037)	(1–265)	
Smelting area	708	18,242	71	1289	49	20,359
	(26–2529)	(3845–52,830)	(4–245)	(40–9054)	(2–244)	
Living area	30	10,240	12	132	44	10,457
	(22–37)	(1320–26,700)	(10–14)	(65–260)	(6–98)	
Average	381	12,669	38	853	35	13,976
	(15–2529)	(355–81,500)	(4–245)	(7–9054)	(0.9–265)	

^a^ ER refers to the ecological risk.

**Table 4 ijerph-19-09698-t004:** Health risks of different land-use types in soils within 2 km of the smelteries.

Land Use	CR ^a^-As(×10^−6^)	NCR ^b^-Cd	NCR-Cu	NCR-Pb	NCR-Zn	Total NCR
Forest land	126.7	0.3	0.1	22.0	0.02	22.3
	(35.3–273.5)	(0.04–1.2)	(0.01–0.2)	(0.5–63.2)	(0.003–0.07)	(0.6–63.4)
Farmland	270.9	0.3	0.03	6.2	0.1	6.5
	(12.8–1891.2)	(0.01–2.3)	(0.01–0.2)	(0.1–50.5)	(0.01–0.9)	(0.1–50.5)
Smelting area	514.5	0.5	0.1	16.1	0.2	16.8
	(22.1–2120.6)	(0.1–1.5)	(0.01–0.4)	(0.5–113.2)	(0.02–0.8)	(0.5–113.5)
Living area	25.1	0.3	0.02	1.7	0.1	2.0
	(19.1–31.1)	(0.04–0.8)	(0.016–0.024)	(0.8–3.3)	(0.02–0.3)	(0.9–4.3)
Average	1701.3	0.3	0.1	22.0	0.02	11.9
	(12.8–2120.6)	(0.01–2.3)	(0.01–0.4)	(0.05–57.0)	(0.003–0.9)	(0.1–113.5)

^a^ CR refers to the carcinogenic risk; ^b^ NCR refers to the non-carcinogenic risk.

## Data Availability

Not applicable.

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
