# Peer review of "Pollution and Risk Assessments of Heavy Metal(loid)s in the Soil around Lead-Zinc Smelteries via Data Integration Analysis"

_ijerph, 2022, doi:10.3390/ijerph19159698_

Round 1

Reviewer 1 Report

This work is extremely ambitious, deals with a topic of relevant interest worldwide.
However, for my perspective it lacks some aspects that should be revised for publication in this journal.
As regards the "Data collection" need to clarified the selection criteria for the screening of literature (lines 90-94).
Also, nearly 50% of the sites in table S1 are in China, this could lead to incorrect evaluations as the study is carried out on a planetary scale.
The same Table S1 shows 5 categories of sampling sites and not 4 as indicated in line 95.
The reference [27] at line 100 is incorrect, the date in 1979 not 1969. This study refers to sediments and not soils; same for the reference [30] (line 112).
The reference [31] (line 131) instead considers only the "Sb", not the contaminants of your work.
The section 2.5 "Data analysis" is not clear, it should be written better reporting at least the main steps and algorithms used.
It is unclear why, the dataset was divided into two groups according to the distances reported in lines 158-162.
This could lead to an underestimation of the problem at greater distances from the smelteries.
In fact, the reference [6] reports in the conclusions that metal(loid)s can often be recognized tens of km away from the source.
The conclusions should be rewritten on the basis of the above and be more consistent.

Author Response

Responses to review’s comments

Reviewer 1This work is extremely ambitious, deals with a topic of relevant interest worldwide. However, for my perspective it lacks some aspects that should be revised for publication in this journal.
1、As regards the "Data collection" need to clarified the selection criteria for the screening of literature (lines 90-94).

Reply: Thanks for your suggestion. We have clarified the selection criteria in the "Data collection" section. All the articles were searched from peer-viewed literature databases, such as Web of Science, Science Direct, and China National Knowledge Infrastructure (CNKI). The articles were further screened according to several criteria, including the number of sampling sites larger than 3, soil sampling depth less than 30 cm, sampling radius less than 20 km from the smelteries, and provided the statistical values of heavy metal(loid) concentrations.

2、Also, nearly 50% of the sites in table S1 are in China, this could lead to incorrect evaluations as the study is carried out on a planetary scale.

Reply: Thanks for your suggestion. After searching the literature according to the keywords, we did find that most of the research regarding soil pollution from lead and zinc smelters was conducted in China. China is the largest producer of Pb and Zn in the world. In 2018, China's refined Pb and Zn output was 483×104 t and 573×104 t, accounting for 41.5% and 43.1% of the total global production, respectively.

In the past 20 years, the Chinese government has paid more attention to environmental protection, and the related investment has also increased. Therefore, there are many related studies. This study does not attempt to elucidate all conclusions on a global scale, we focus on the effects of land use types and shed light on the reasons for such effects in different countries.

3、The same Table S1 shows 5 categories of sampling sites and not 4 as indicated in line 95.

Reply: Thanks for your suggestion. A few literatures did not provide information on land use types,

The “5” in Table S1 refers to the land use type not mentioned. We have added this description to the table footnote.

4、The reference [27] at line 100 is incorrect, the date in 1979 not 1969. This study refers to sediments and not soils; same for the reference [30] (line 112).

Reply: Thanks for your suggestion. We have changed the reference [27] and [30]. The potential ecological risk (PER) index, proposed by Hakanson (1980), was introduced to assess the degree of heavy metal pollution in soils. Although this method is based on the principle of sedimentology and aquatic ecosystem, but it has been used in the soil pollution evaluation and got a certain reference value (Shen et al, 2017).

Shen, F., Liao, R.M., Ali, A., et al., Distribution and Risk Assessment of Heavy Metals in Soil Near a Pb/Zn Smelter in Feng County. Ecotoxicol. Environ. Saf., China, 2017. 139:p. 254-262.

5、The reference [31] (line 131) instead considers only the "Sb", not the contaminants of your work.

Reply: Thanks for your suggestion. We have changed the reference [31] here.

6、The section 2.5 "Data analysis" is not clear, it should be written better reporting at least the main steps and algorithms used.

Reply: Thanks for your suggestion. We have added details of the main steps and algorithm.

7、It is unclear why, the dataset was divided into two groups according to the distances reported in lines 158-162. This could lead to an underestimation of the problem at greater distances from the smelteries.

Reply: Thanks for your suggestion. According to Fig.1, we found that distance has a large impact on the metal concentrations in soil. To study the impact of land use types on soil pollution, we need to reduce the effects of the sampling radius between studies. Therefore, we used the data within 2 km radius of Pb-Zn smelteries.

8、In fact, the reference [6] reports in the conclusions that metal(loid)s can often be recognized tens of km away from the source.

Reply: Thanks for your suggestion. The emissions from smelteries do affect the heavy metal concentrations in soil tens of km away. However, the mean concentrations of heavy metal(loid)s in the soils within 2 km to smelteries were significantly higher than those outside this distance, suggesting the soil pollution is most serious within 2 kilometers of the source.

9、The conclusions should be rewritten on the basis of the above and be more consistent.

Reply: Thanks for your suggestion. The conclusion is revised to be more precise.

Reviewer 2 Report

The manuscript by Zhou et al., is focused on pollution, potential ecological risk, and health risk of soils around Pb-Zn smelteries globally. Data integration analysis were conduct through 54 peer-reviewed reports. Although the analysis methods are traditional, the data are sufficient, representative, and globally. Overall, the results conclude As, Cd, Pb and Zn as the priority pollutions for protecting ecosystem and human health, and suggest risk of soils within 2km of Pb-Zn smelteries should be of concern. Those results might guide for risk assessment in the future.

The manuscript suffers from some minor defects regarding the obtained results. In my opinion the paper can be published after minor revision.

1.A suggestion: the categories of Group#1 and Group#2 described in line 159-160 should be placed in 2.1 Data collection.

2.Line 103: “Bi values were set to the upper continental crust values (mg/kg) in the current study, which are 5.7, 0.06, 27, 25, and 75 for As, Cd, Cu, Pb, and Zn, respectively.”

Whether the Bi value should be different in different countries or places? Using the upper value may reduce the overall value.

3. Line 108: the full name of PI (Pollution index) was described in line 52 before, so abbreviation can be used directly.

4.Line 175: the units of mean concentration are missing.

Line 293: The units of standard value of soil environment are also missing in Table 2.

5. As described in Line 244, Line 298 and Line 317, the analyzed data were Group#1(within 2 km to smelteries). Thus, “soil near Pb-Zn smelteries” in the titles of Fig.3, Table 2, Table 3, and Table 4 should be defined more clearly.

6. Line 259: The Y axis title in the right of Fig.3 should be added.

Author Response

Responses to review’s comments

Reviewer 2The manuscript suffers from some minor defects regarding the obtained results. In my opinion the paper can be published after minor revision.

1.A suggestion: the categories of Group#1 and Group#2 described in line 159-160 should be placed in 2.1 Data collection.

Reply: Thanks for your suggestion. Comment followed.

2.Line 103: “Bi values were set to the upper continental crust values (mg/kg) in the current study, which are 5.7, 0.06, 27, 25, and 75 for As, Cd, Cu, Pb, and Zn, respectively.”

Whether the Bi value should be different in different countries or places? Using the upper value may reduce the overall value.

Reply: Thanks for your suggestion. The background values of heavy metals in soil are spatially varied. However, the background values in some of the countries are unavailable. In this case, many studies have adopted the upper continental crust (UCC) values to substitute the background value, e.g. Karande et al, 2020. Hence, we also adopted the UCC as the reference values of heavy metal(loid)s.

3.Line 108: the full name of PI (Pollution index) was described in line 52 before, so abbreviation can be used directly.

Reply: Thanks for your suggestion. Comment followed.

4.Line 175: the units of mean concentration are missing.

Reply: Thanks for your suggestion. Comment followed.

5.Line 293: The units of standard value of soil environment are also missing in Table 2.

Reply: Thanks for your suggestion. Comment followed.

6.As described in Line 244, Line 298 and Line 317, the analyzed data were Group#1(within 2 km to smelteries). Thus, “soil near Pb-Zn smelteries” in the titles of Fig.3, Table 2, Table 3, and Table 4 should be defined more clearly.

Reply: Thanks for your suggestion.We have revised these titles.

7.Line 259: The Y axis title in the right of Fig.3 should be added.

Reply: Thanks for your suggestion. We have added the Y axis title in the revised manuscript.

Reviewer 3 Report

This is a standard, mostly technical, paper. I have nothing against publication. However, I would see some moderate revisions aimed at improving some key aspects and dimensions of the study:

1) representativeness of the study area in a broader research context should be clarified.

2) Technical details and literature review should be expanded significantly, eventually providing a specific appendix at the end of the text.

3) novelty of the present procedure in comparison with earlier studies can be clarified a bit more.

4) What is the contribution of this study for scholars outside China? You specifically refer to Europe, that's fine. Any further information on that?

Author Response

Responses to review’s comments

Reviewer 3:This is a standard, mostly technical, paper. I have nothing against publication. However, I would see some moderate revisions aimed at improving some key aspects and dimensions of the study:

1.Representativeness of the study area in a broader research context should be clarified.

Reply: Thanks for your suggestion. We hoped this study would provide suggestions for pollution identification and risk prevention of heavy metals in smelting-affected areas.

2.Technical details and literature review should be expanded significantly, eventually providing a specific appendix at the end of the text.

Reply: Thanks for your suggestion. We have added the Screening criteria for the literature collection in section 2.1. and also provided detailed information on the collected literature in the “Supporting information” as an appendix (table S1)。

3.Novelty of the present procedure in comparison with earlier studies can be clarified a bit more.

Reply: Thanks for your suggestion. This work studied the data of heavy metals around the smelteries globally and compared them with SSVs from different countries. Also, we studied the impacts of land use on soil pollution around the smelteries. The introduction part is revised accordingly.

4.What is the contribution of this study for scholars outside China? You specifically refer to Europe, that's fine. Any further information on that?

Reply: Thanks for your suggestion. This study does not attempt to elucidate all conclusions on a global scale. We focus on the impact of land use on soil pollution around the Pb/Zn smelteries and reveal the effects of SSVs from different countries on risk identification.

Round 2

Reviewer 1 Report

The authors have addressed my observations. However, I'll have to disagree with them about the following concerns.

In my opinion, "global scale" is not a right keyword for this work. In fact, the authors respond to my earlier commens that "this study does not attempt to elucidate all conclusions on a global scale, we focus on the effects of land use types and shed light on the reasons for such effects in different countries".

In my opinion, the sampling radius of 2 km and 20 km are not enough to deal with this issue. The difference is substantial because the transport and diffusion of contaminants do not have the same incidence, for example, at 5 km from the smeltery rather than 18 km. This could lead to an underestimation of the issue at distances between the two radius considered.

Author Response

The authors have addressed my observations. However, I'll have to disagree with them about the following concerns. In my opinion, "global scale" is not a right keyword for this work. In fact, the authors respond to my earlier commens that "this study does not attempt to elucidate all conclusions on a global scale, we focus on the effects of land use types and shed light on the reasons for such effects in different countries".

Response: thank you for the suggestion. We have deleted the “global scale” from the keyword.

In my opinion, the sampling radius of 2 km and 20 km are not enough to deal with this issue. The difference is substantial because the transport and diffusion of contaminants do not have the same incidence, for example, at 5 km from the smeltery rather than 18 km. This could lead to an underestimation of the issue at distances between the two radius considered.

Response: Thanks for the comment. We adopted 2km to divide the dataset because most of the studies focus on soil pollution within the distance of 0 – 2 km to the smelteries.

We added a new figure S1 to the supplemental file as attached: We found that the sampling radius in most of the studies was between 0 – 2 km to the smelteries. Concentrations and NIPI of heavy metals in the soils with a sampling radius of less than 2 km were much higher than that between 2 km and 20 km.

Reviewer 3 Report

The revision was carried out diligently but I would see some more and convincing explainations for issue 1

1.Representativeness of the study area in a broader research context should be clarified.

Reply: Thanks for your suggestion. We hoped this study would provide suggestions for pollution identification and risk prevention of heavy metals in smelting-affected areas.

The reply is not totally convincing because the textual revisions are relatively short and possibly not completely point. Can you enrich them?

Author Response

The revision was carried out diligently but I would see some more and convincing explainations for issue 1

 1.Representativeness of the study area in a broader research context should be clarified.

Reply: Thanks for your suggestion. We hoped this study would provide suggestions for pollution identification and risk prevention of heavy metals in smelting-affected areas.

 The reply is not totally convincing because the textual revisions are relatively short and possibly not completely point. Can you enrich them?

Response: Thank you for the comment. The Pb-Zn smelting industry is one primary source of heavy metals, which often leads to severe soil pollution in surrounding areas. The large number of Pb-Zn smelting sites and the high level of soil pollution makes soil remediation and risk management in the areas very difficult.

To study the distribution patterns of heavy metals around Pb-Zn smelting sites, we collected data from different peer-viewed literature databases, including both English and Chinese databases. These studies cover all continents and 18 countries around the world. At the same time, we considered the differences in sampling depth, sampling radius and the land use type of sampling sites among the studies. After data collection, statistical analyses and risk assessments were performed without bias. It is not a case study for a specific site, but a comprehensive analysis including data from dozens of smelting sites, so we consider the study area to be representative.

This manuscript is a resubmission of an earlier submission. The following is a list of the peer review reports and author responses from that submission.